# Ultraviolet-Ozone Treatment: An Effective Method for Fine-Tuning Optical and Electrical Properties of Suspended and Substrate-Supported MoS_2_

**DOI:** 10.3390/nano13233034

**Published:** 2023-11-27

**Authors:** Fahrettin Sarcan, Alex J. Armstrong, Yusuf K. Bostan, Esra Kus, Keith P. McKenna, Ayse Erol, Yue Wang

**Affiliations:** 1School of Physics, Engineering and Technology, University of York, Heslington, York YO10 5DD, UK; alex.armstrong@york.ac.uk (A.J.A.); keith.mckenna@york.ac.uk (K.P.M.); 2Department of Physics, Faculty of Science, Istanbul University, Vezneciler, Istanbul 34134, Turkey; yusufkerembostan@gmail.com (Y.K.B.); esraakus2021@gmail.com (E.K.); ayseerol@istanbul.edu.tr (A.E.); 3Institut d’Electronique, Microelectronique & Nanotechnologie IEMN CNRS UMR 8520, Université Polytechnique Hauts de France, 59313 Valenciennes, France

**Keywords:** MoS_2_, ultraviolet-ozone (UV-O_3_), doping, surface treatment, transition metal dichalcogenide, field-effect transistor, photoluminescence, density functional theory (DFT)

## Abstract

Ultraviolet-ozone (UV-O_3_) treatment is a simple but effective technique for surface cleaning, surface sterilization, doping, and oxidation, and is applicable to a wide range of materials. In this study, we investigated how UV-O_3_ treatment affects the optical and electrical properties of molybdenum disulfide (MoS_2_), with and without the presence of a dielectric substrate. We performed detailed photoluminescence (PL) measurements on 1–7 layers of MoS_2_ with up to 8 min of UV-O_3_ exposure. Density functional theory (DFT) calculations were carried out to provide insight into oxygen-MoS_2_ interaction mechanisms. Our results showed that the influence of UV-O_3_ treatment on PL depends on whether the substrate is present, as well as the number of layers. Additionally, 4 min of UV-O_3_ treatment was found to be optimal to produce p-type MoS_2_, while maintaining above 80% of the PL intensity and the emission wavelength, compared to pristine flakes (intrinsically n-type). UV-O_3_ treatment for more than 6 min not only caused a reduction in the electron density but also deteriorated the hole-dominated transport. It is revealed that the substrate plays a critical role in the manipulation of the electrical and optical properties of MoS_2_, which should be considered in future device fabrication and applications.

## 1. Introduction

In recent years, semiconducting two-dimensional transition metal dichalcogenides (2D-TMDs) have become a group of highly demanded materials for next-generation optoelectronic devices such as photodetectors and light emitters [1,2,3,4] because of their unique optical, electronic, and structural properties. Despite many advantages of these semiconducting materials, there are some drawbacks such as low carrier concentration and mobility, which result in low electrical conductivity [5,6] compared to the materials already widely used in electronic/optoelectronic technologies, such as Si and GaAs. To realize high-performance optoelectronic devices, both n-type and p-type semiconducting 2D-TMDs are required [7,8]. Conventional doping techniques used for semiconductors are not suitable for 2D materials because they modify their crystal structures and result in a significant deterioration of their optoelectrical properties [9]. On the other hand, thanks to the atomic thickness of 2D-TMDs, their optical and structural properties as well as carrier dynamics can be efficiently engineered using different post-growth methods [10]. Surface charge transfer-based doping [11], substitutional doping [12], interstitial doping [13], and vacancy-based doping [14] are the main post-growth doping mechanisms for 2D-TMDs [15]. Based on these mechanisms, there are several reported doping techniques such as chemical treatment [13], ion implantation [16], plasma doping [17], thermal annealing [18], electron beam irradiation [19], and ultraviolet-ozone (UV-O_3_) treatment [20], which aim to achieve n- and/or p-type doping. Due to the high sensitivity of 2D materials’ properties, the main challenge in post-growth doping is to maintain their superb optoelectrical properties for device applications, while controlling the doping concentration with consistency. In this study, we focus on the effect of UV-O_3_ treatment on the optical and electrical properties of MoS_2_, which leads to p-type doping of the TMD material.

UV-O_3_ treatment is a useful process for a wide range of materials for surface modification [21]. It has been employed as an effective tool for defect engineering and doping in graphene and 2D TMDs. There are a few studies in the literature on UV-O_3_-induced p-type doping on graphene [22,23,24]. Liang et al. presented controllable p-type doping in a range of semiconducting TMDs (MoTe_2_, WSe_2_, MoSe_2_, and PtSe_2_) and proposed three mechanisms for UV-O_3_-induced hole doping: (1) charge transfer due to the interaction with oxygen molecules, (2) isoelectronic substitution of chalcogen atoms with oxygen atoms, and (3) charge transport over the oxide surface due to the transition metal oxide formation (MoO_3_, WO_3_ etc.) [20]. Zheng et al. showed that UV-O_3_ treatment is an effective method for p-type doping of MoTe_2_ field-effect transistors and it enhances its electrical performance enormously. The hole concentration and mobility are enhanced by nearly two orders of magnitude, and the conductivity by five orders of magnitude [25]. These promising studies are focused on the effect of UV-O_3_ treatment on the electrical properties of TMDs only. In 2D-TMD optoelectronic devices, the effect of UV-O_3_ treatment on their optical properties is also critical. There are only a few studies about the effect of UV-O_3_ treatment on the optical properties of semiconducting TMDs. Yang et al. reported that PL intensity of pristine exfoliated MoS_2_ decreased and was eventually quenched as UV-O_3_ exposure time increased from 0 to 10 min, which was attributed to the structural degradation [26]. The quenching of PL was also observed on chemical vapour deposited and exfoliated monolayer MoS_2_ [27] and on exfoliated monolayer WS_2_ and WSe_2_ as a result of 6 min of UV-O_3_ treatment due to oxidation [28]. On the other hand, Zheng et al. showed a 37-fold increment in the PL intensity by converting trilayer MoSe_2_ to monolayer with 7 min of UV-O_3_ treatment [29]. While the effect of UV-O_3_ on the electrical properties is consistent from one study to another, there is a contradiction about the effect of UV-O_3_ treatment on the optical properties of 2D-TMDs. More importantly, there is no systematic study on the effect of UV-O_3_ treatment on suspended, or multi-layer TMDs. Although monolayer (ML) TMDs have the strongest emission, the effect on few-layer TMDs is also important for the potential electronic and optoelectronic devices.

In this paper, we reveal that the effect of UV-O_3_ treatment does not only depend on the type of materials but also the number of layers and the substrate (the environment). To investigate the layer and substrate dependency of the UV-O_3_ treatment on the optical properties of semiconducting TMDs, we fabricated suspended and substrate-supported MoS_2_ samples with different numbers of layers from monolayer to seven layers. PL spectroscopy was carried out to investigate the effect on their optical properties with a range of UV-O_3_ exposure time. Density functional theory (DFT) calculations were performed to understand which oxygen-related mechanism during the UV-O_3_ exposure could cause p-type doping via charge trapping, as well as the role of intrinsic sulphur defects. Further electrical characterisation was performed on a 4L-MoS_2_ transistor to investigate how the carrier dynamics can be modified using the UV-O_3_ treatment.

## 2. Experimental Methods

E-beam lithography was used to pattern the squares (4 μm × 4 μm) on Polymethyl methacrylate (PMMA)-coated Si_3_N_4_ (150 nm)-on-SiO_2_ substrates. Patterned squares were etched 300 nm in depth with a Reactive Ion Etcher, using a mixture gas of CHF_3_ and O_2_. Resist was removed with a resist remover (1165) and acetone, and the substrate was rinsed in isopropyl alcohol (IPA) before drying. Bulk single-crystal MoS_2_ was purchased from HQ Graphene, which was intrinsically n-doped. Scotch tape and polydimethylsiloxane (PDMS)-assisted mechanical exfoliation method was used to obtain MoS_2_ flakes with different numbers of layers. The flakes larger than 10 μm × 10 μm were transferred either onto the etched squares to be suspended or the flat area of the Si_3_N_4_-on-SiO_2_ substrates.

The PL spectra of the flakes were measured using a microPL setup equipped with an Andor iDus detector and a 532 nm excitation laser, before and after UV-O_3_ treatment. The laser beam was focused to be a spot of ~1.5 μm in diameter, fitting inside the suspended areas. The UV-O_3_ treatment was performed (Jelight, 30 mW/cm^2^ at a wavelength of 254 nm) for different periods of time. 

DFT calculations were undertaken on four-layer MoS_2_ using a plane wave basis set, as implemented in the VASP package [30,31,32]. A gamma-centred Monkhorst-Pack grid of 2 × 2 × 1 *k* points was used to sample the Brillouin zone for all calculations, with geometry optimizations performed to a force tolerance of 0.01 eV/Å. Ideal values of D3 parameters for HSE06 are still an open area of research and could therefore not be sourced from the literature, so in this study, we used parameters quoted for the related hybrid functional PBE0 for calculations. Version 5.2 PBE plane wave potentials (PAW) were used for all calculations, with a plane wave cut-off value of 520 eV.

Preliminary geometry optimizations were undertaken with the Perdew–Burke–Ernzerhof (PBE) functional [33], including Grimme’s D3 van der Waals corrections [34] to determine the lowest energy adsorption site and adsorbate bond orientations. Further calculations of the band structures, including substitutional defects, were undertaken using the HSE06 functional with D3 corrections, ensuring more accurate energy levels to fully capture any charge-trapping levels [35]. These calculations were initially optimized with one extra electron added to capture the geometry of the charged structure, then reoptimized for the neutral case. This enabled Bader charge analysis to be conducted on both neutral and charged cases to identify if the charged case differed greatly at the defect site, which would be indicative of charge trapping. To reduce the defect density and prevent bands forming from the defect levels, the unit cell used was double the size of the primitive cell in the *a* and *b* directions. A vacuum gap of 20 Å was employed to minimize interaction between periodically repeated slabs.

The band structure was also calculated for a pristine four-layer MoS_2_ system at the HSE06 level, with the parameters described above and a high symmetry *k*-point path generated using the sumo package [36]. This was necessary to determine where the defect levels were relative to these bands. We note here that spin-orbit coupling was not included in any of these calculations; however, previous calculations on MoS_2_ have shown this to result in a 0.1 eV shift in the band gap—significantly lower than the energy difference between the CBM and defect levels found in this work—hence it would be unlikely to change any conclusions [37].

Defect formation energies were calculated for each defect as a function of the chemical potential of sulphur to determine which defects were most stable under different conditions—from sulphur-poor to sulphur-rich. These energies were calculated using Equation (1), where *μ_i_* are the chemical potentials of species removed or added to the system and *n_i_* is the change in the number of these atoms, *i* = Mo, S or O.
E_f_^def^ = E^def^ − E^bulk^ − ∑*n_i_μ_i_*(1)

Both sulphur-rich and sulphur-poor limits were considered at a constant oxygen chemical potential—with only 2 points necessary due to the clear linear nature of these energies with respect to one chemical potential. These limits were defined as the sulphur chemical potential, where *μ_S_* = 1/2 E_S_2__ and *μ_Mo_* = E_Mo_ for the sulphur-rich and sulphur-poor cases, respectively, as either of these conditions uniquely defines both *μ_S_* and *μ_Mo_* according to Equation (2).
E_MoS_2__ = *μ_Mo_* + 2*µ_S_*(2)

All energies used were calculated at PBE + D3 level of theory and the formation energies for O_2_ defects were halved to calculate per oxygen atom and ensure these energies are comparable between the different systems.

Electronic states with enhanced localization on oxygen-related defects were identified by examining coefficients in the projection of one-electron wavefunctions onto atoms. To ensure the corresponding band energies were consistent between different supercells, the lowest energy molybdenum core state in the pristine surface was a reference for aligning electronic structures. The densities of states were calculated on a grid of 2000 points and plotted using the sumo package [36].

In order to monitor the doping effect with different UV-O_3_ exposure times, a 4L-MoS_2_ field-effect transistor was fabricated. A 4L MoS_2_ flake was transferred on SiO_2_/Si wafer and e-beam lithography was used to pattern a Transfer Length Method (TLM) structure on the flake. Cr/Au (10 nm/60 nm) were deposited in a thermal evaporator, followed by a lift-off process. Electrical characterisation was carried out using an Agilent B2902A double-channel source-meter unit.

## 3. Results and Discussion

It is well known that the electronic band structure and energy bandgap of TMDs depend on the number of layers. The number of layers in MoS_2_ flakes can be precisely determined using the PL peak wavelength/energy. Flakes with different layer numbers were selectively transferred to the pre-patterned Si_3_N_4_ substrates. Figure 1a,b show a microscope image of the 4L MoS_2_ flake on the Si_3_N_4_ substrate, and a schematic drawing of the sample structure, respectively.

Optical properties of suspended and supported MoS_2_ flakes on Si_3_N_4_ were investigated using micro-PL spectroscopy. As a function of the number of layers (from a single layer to seven layers) the bandgap of the suspended MoS_2_ varies from ~1.90 eV to ~1.35 eV (Figure 1e). Up to 45 meV blueshifts in the PL peak energy were observed in the thicker supported flakes compared to the suspended flakes [38]. There are several studies on the substrate effect on the optical properties of 2D TMDs in the literature [39,40,41], although most of these studies were focused on the substrate effect on monolayer TMDs. The weaker, broader, and red-shifted PL from the TMDs transferred onto dielectric substrates can be attributed to the amount of moisture at the interface and/or interlayer charge transfer, which results in charge doping [41,42,43]. In Figure 1, we observe a consistent reduction in PL intensity for all layer numbers, and a small redshift of the peak energy in the thicker supported flakes compared to the suspended flakes. This can be explained by the possibility of more charge transfer in the thicker layers due to their lower energy bandgap [44].

Next, the effect of UV-O_3_ treatment on suspended and supported MoS_2_ as a function of number of layers and exposure time was investigated. The main purpose of our study is to understand the layer and substrate dependency of the UV-O_3_ treatment time on the optical properties of the 2D TMDs, ideally maintaining their photoluminescence. We chose the UV-O_3_ treatment times of 1, 4, 6, and 8 min on each sample. Each sample was exposed separately. Figure 2 shows the PL spectra of pristine and UV-O_3_-treated flakes. As UV-O_3_ treatment time increases, the PL intensity decreases for all samples. The intensity of the PL emission from the supported MoS_2_ decreases significantly with UV-O_3_ treatment time and is eventually fully quenched after 8 min., while the suspended flakes are less affected by the treatment. Figure 3 shows that there are clear differences in the trend between the suspended and supported flakes. In general, the thinner flakes are more sensitive to UV-O_3_ treatment for both cases, which is expected; however, for suspended flakes, up to 4 min of exposure for all thicknesses does not degrade the PL intensity significantly. We note that in some layers, the PL intensities after 4 min treatment are even higher than after 1 min treatment time, which is likely due to the small variation in PL intensities between consecutive measurements. The overall trend of above 4 min exposure time is, however, clear.

We use DFT calculations to understand the effect of UV-O_3_ on suspended TMDs. Conventional UV-O_3_ cleaners have two dominant UV peaks, at 184 nm and 254 nm. Upon irradiation, molecular oxygen (O_2_) present in the air is dissociated by radiation at 184 nm. This results in the formation of two radicals of singlet oxygen (O). These radicals continue to react with molecular oxygens forming molecules of ozone (O_3_). The simulations were carried out for these three species. Sulphur substitutional defects involving these species, as well as surface adsorption onto various non-symmetrically equivalent high symmetry sites, were considered. Both substitution and adsorption of O_3_ resulted in dissociation into an O defect and a free O_2_ molecule in the vacuum gap, which is therefore equivalent to the O defects. The formation energies were calculated with a constant oxygen chemical potential of half the energy of an oxygen atom, plotted in Figure 4a, which represents the stability of each type of defect as a function of sulphur chemical potential. In both the sulphur-rich and sulphur-poor extremes, no O_2_ defects were stable as their formation energies were above zero for all chemical potentials, hence O defects can be concluded to be the most likely defect to form in this system. In the sulphur-poor region, O_2_/S substitution is the most stable, and for all chemical potentials, O adsorption is the most stable scenario. Figure 4b shows that the unoccupied electronic states introduced by all these defects are higher in energy than the conduction band minimum (CBM) of pristine monolayer MoS_2_, meaning no PL shift should be induced by the defects. The O_2_ defects have higher formation energy than the O defects, which indicates that they are likely to be short-lived; however, these short-lived O_2_ defects could be relevant for providing mechanisms to form O defects. In Figure 4c, the excess Bader charge was shown as a function of the vertical position of each atom in the unit cell in the case of O adsorption and with one added electron. Low localization can be seen at both top and bottom surfaces, with the most localization on the sulphur atom closest to the adsorbed O atom. Even this larger localization is only 0.08 e, suggesting this localization is extremely weak and unlikely to trap enough charge to cause p-type doping. One singlet O adsorption between the TMD layers was also investigated at the PBE level (the lower level of theory) for both neutral and +1 electron systems. This gave similar charge densities and defect levels compared to O adsorption on the surface. We also note here that our DFT model reflects the substitution of one sulphur atom in the unit cell (Figure 4c), and the adsorption of one of the three oxygen species (O, O_2_, and O_3_) in the suspended 4L MoS_2_ system; it does not reflect multiple species adsorption or substitution of multiple sulphur atoms, which potentially happens with longer UV-O_3_ treatment time. The PL results from short-time UV-O_3_ treatment (1 to 4 min) show no significant reduction in the intensity nor any peak wavelength shift, which is consistent with our DFT results.

Inclusion of the substrate in the DFT calculations is too computationally time-consuming using the hybrid level of theory and requires taking into account extra parameters, such as surface roughness, optical interferences specifically on dielectric substrates, defects, and impurities on the interface, etc., which is considered to be future work.

To understand the effect of UV-O_3_ on the substrate-supported flakes, we analyzed the PL spectra of the 2L and 6L flakes in depth. The reduction in the PL intensities starts from higher energy on the supported 2L and 6L flakes with up to 6 min of UV-O_3_ treatment (see Figure 2a,c). This can be explained by carrier dynamics involving the presence of the Si_3_N_4_ substrate. The bandgap and photogenerated charge carriers of the substrate under UV irritation play a key role in this process. The bandgap of the Si_3_N_4_ is ~5 eV. During the UV-O_3_ exposure, the 184 nm characteristic radiation of the UV source can generate free electrons and holes in the substrate. The generated free electrons diffuse to MoS_2_, which has a lower energy level than that of the conduction band of Si_3_N_4_ in the heterostructure (illustrated in Figure 5a). Such diffusion compensates for the decreased electron density caused by the oxygen adsorption and allows more oxygen defect formation on the MoS_2_ surface. The transferred electrons also recombine with free holes in MoS_2_, resulting in non-radiative recombination at the interfaces of the flake and the substrate, hence further reduction in the PL intensity of the supported MoS_2_. Moreover, we observed an 11 meV redshift in the PL peak energy of the 4L supported flakes, compared to the suspended 4L flakes, which supports the substrate-induced charge carrier dynamics interpretation. Such substrate-dependent carrier dynamics have been observed previously in graphene [24].

Finally, we fabricated a substrate-supported field effect transistor with a TLM structure using 4L MoS_2_ to investigate the doping effect of the UV-O_3_ treatment. The optical microscope image and illustrations of the fabricated device are presented in Figure 6a. The structure includes five transistors with different channel lengths from 2 μm to 4 μm in 0.5 μm step, and the consecutive two electrodes can be used as source and drain. The channel width of the transistors is 12 μm. All transistors (with different channel lengths) exhibit the same trend of the input and output characteristics against UV-O_3_ treatment. We present the 4 μm length device in detail, in Figure 6b–d, as a function of UV-O_3_ treatment time. Figure 6b shows the input characteristic of the transistor.

The pristine MoS_2_ transistor shows n-type dominant ambipolar characteristics; with increasing the UV-O_3_ treatment time, the n-type dominancy reduces and eventually, the characteristic changes from electron dominant to hole dominant, as seen in Figure 6b,c. With 6 min treatment time, the on/off ratio for the n-type dominant reduced from 10^4^ to zero, while that of the p-type dominant increased from 1 to 10^3^ under 0.5 V bias. On the other hand, when above 6 min of UV-O_3_ treatment, not only is the electron density reduced but so is the hole density, and the drain-source current is two orders of magnitude lower after 8- and 10-min of treatment (Figure 6d).

## 4. Conclusions

The effect of UV-O_3_ treatment on the optical and electrical properties of mono- and multi-layer MoS_2_ is presented. The PL intensity from substrate-supported MoS_2_ decreases significantly with UV-O_3_ treatment time and is fully quenched after 8 min of treatment. The PL of suspended flakes is, however, less affected by the UV-O_3_ treatment. Our DFT results suggest that no significant charge trapping or PL peak energy shift should be expected from the substitution and adsorption of O, O_2_, and O_3_ on the surface of a suspended MoS_2_ flake. We conclude that the substrate plays a critical role in the UV-O_3_-induced manipulation of the electrical and optical properties of MoS_2_. The effect of UV-O_3_ treatment is also layer-thickness dependent; while the thinner flakes could experience a 90% reduction in the PL intensity, the thicker suspended flakes remain above 60% of their original intensity. Our electrical measurements show that above 6 min of UV-O_3_ treatment not only causes a reduction in the electron density but it also deteriorates the hole transport. Our work suggests that 4 min UV-O_3_ treatment is the optimum time to produce p-type MoS_2_ and maintain above 80% of PL intensity without any shift in emission wavelength. Longer than 4 min leads to deterioration of both optical and electrical properties, which should be taken into consideration in future device fabrication processes.

## Figures and Tables

**Figure 1 nanomaterials-13-03034-f001:**
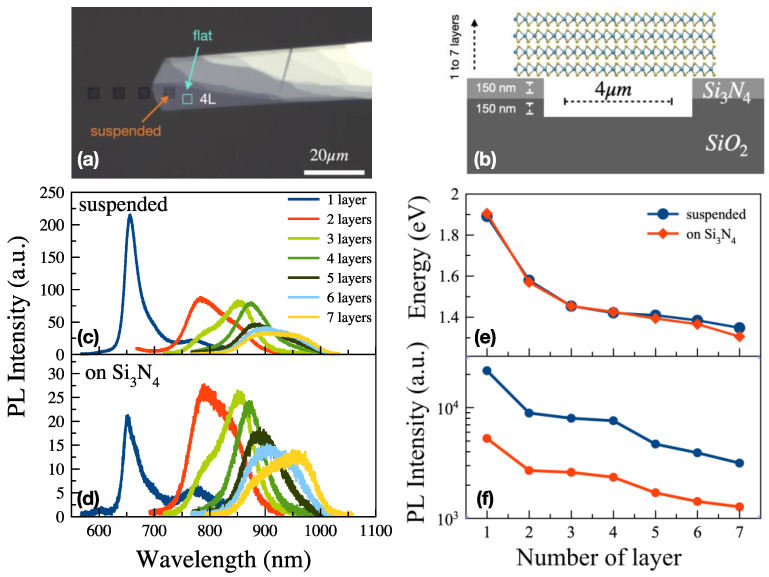
(**a**) Microscope image of the 4L MoS_2_ on Si_3_N_4_, (**b**) the schematic of the sample, PL spectra of (**c**) suspended and (**d**) supported MoS_2_ on Si_3_N_4_, and (**e**) PL peak energy and (**f**) intensity as a function of layer numbers.

**Figure 2 nanomaterials-13-03034-f002:**
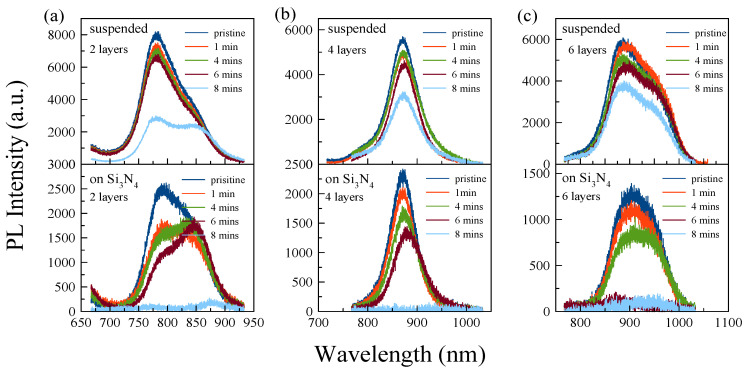
PL spectra of (**a**) two-layer, (**b**) four-layer, and (**c**) six-layer MoS_2_ flakes with different UV-O_3_ treatment times: 1, 4, 6, 8 min.

**Figure 3 nanomaterials-13-03034-f003:**
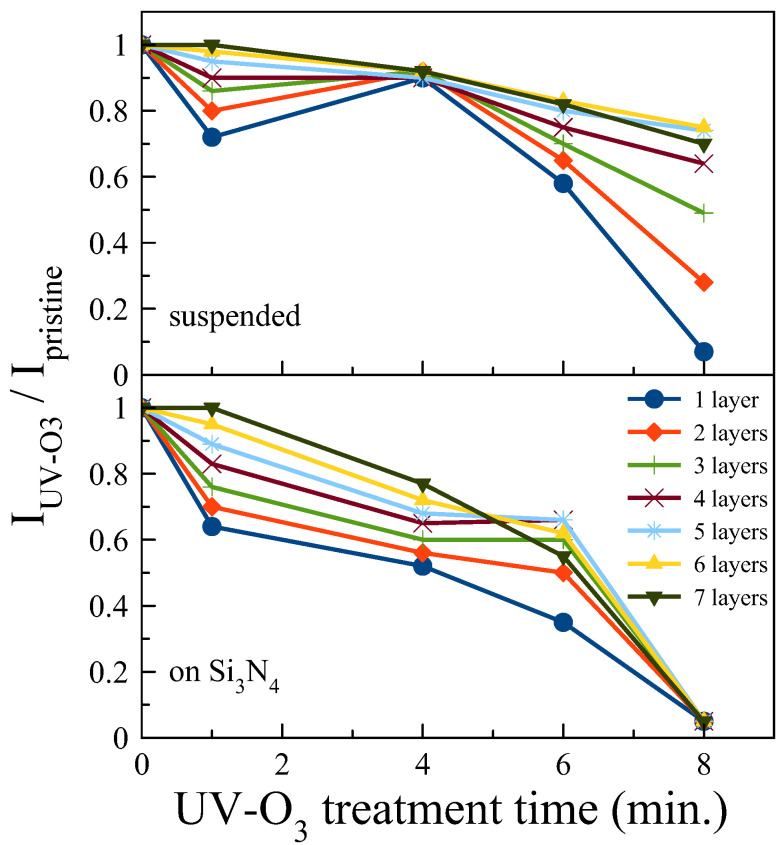
PL intensity ratio as a function of UV-O_3_ exposure time.

**Figure 4 nanomaterials-13-03034-f004:**
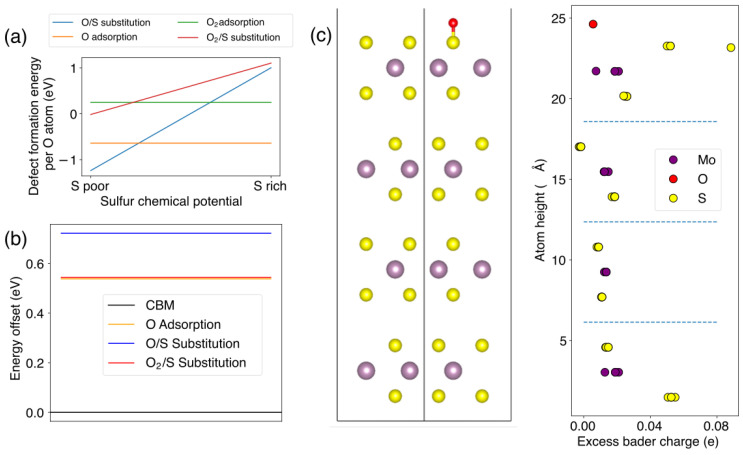
(**a**) Defect formation energies for O (yellow line) and O_2_ (green line) adsorption and S (blue and red lines) substitutional defects as a function of sulphur chemical potential; (**b**) defect energy level offsets for each stable defect mechanism, with respect to the conduction band minimum (CBM, black line) in the pristine 4L-MoS_2_ system; (**c**) excess Bader charge for each atomic site (right), with sites aligned vertically with unit cell (left) for the O adsorption case with the presence of one added electron.

**Figure 5 nanomaterials-13-03034-f005:**
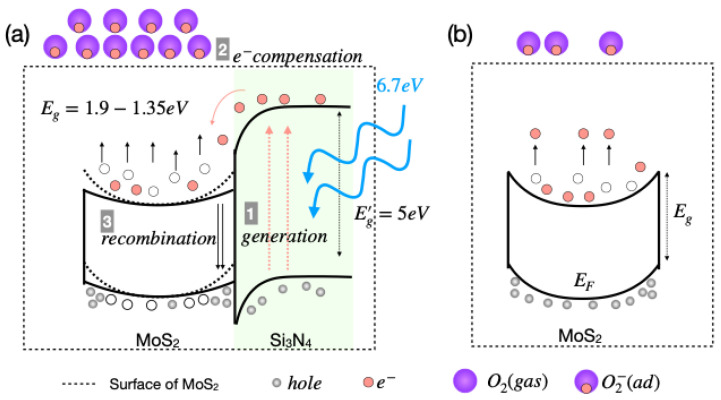
Illustrations of UV-O_3_ treatment effect on (**a**) supported and (**b**) suspended MoS_2_.

**Figure 6 nanomaterials-13-03034-f006:**
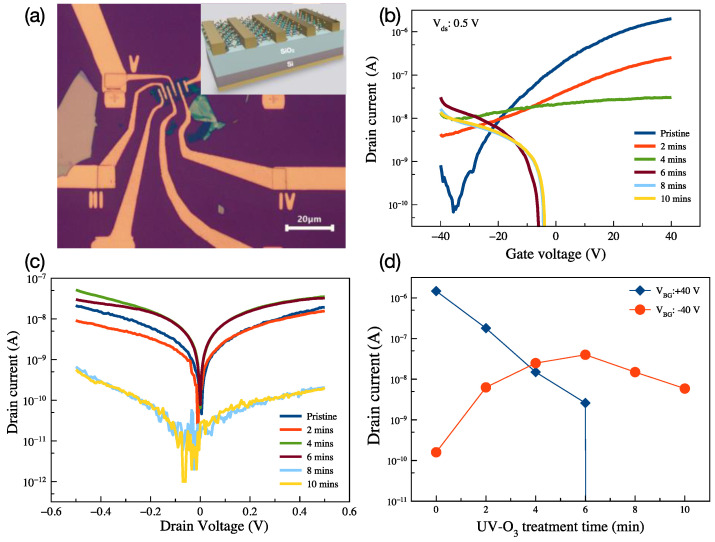
(**a**) Optical microscope image with 20 μm of a scale bar and illustration (inserted) of the MoS_2_ FET, drain-source current as a function of (**b**) applied gate voltage, (**c**) applied drain-source voltage, and (**d**) UV-O_3_ treatment time.

## Data Availability

The authors declare that all the data and code supporting the findings of this study are available within the article, or upon request from the corresponding author.

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
