# Peer review of "Ultraviolet-Ozone Treatment: An Effective Method for Fine-Tuning Optical and Electrical Properties of Suspended and Substrate-Supported MoS2"

_nanomaterials, 2023, doi:10.3390/nano13233034_

Round 1

Reviewer 1 Report

Comments and Suggestions for Authors

Dear Authors,

Your paper submitted to Nanomaterials and entitled "Ultraviolet-ozone treatment: an effective method for fine-tuning optical and electrical properties of suspended and substrate-supported MoS2" is of interest and is worth publishing after some minor revisions.

Below, you can find my comments:

1) In the introduction section, the connection between the state of the art and main purposes of the work is described quite weak. I suggest to improve the relationship between these two parts of the Introduction.

2) Please explain why you took such time intervals for UV-O3 treatment: 2,4,6,8, 10 minutes. What is the scientific sence of these treatment time intevals. I didn't find any information in the manuscript.

3) In Fig.2, the colors of the lines in the legends are very faded, so it is difficult to make out which curves they belong to. I recommend making the colors of the lines in the legends more vivid.

4) In Fig. 2b, it looks that the scales of the Si3N4 and SiO2 layer thicknesses (150 nm each) are not proportional to the scale of the pattern width (4 microns). I understand that this is a schematic, but it gives the reader the wrong impression at first glance. Please try to solve this problem.

5) In Fig 6, there is an error in the graph letter indexing:  "UV-O3 treatment time " corresponds to the d) and "applied drain-source voltage" corresponds to the c). You typed an inverse version.

Author Response

Reviewer 1.

Your paper submitted to Nanomaterials and entitled "Ultraviolet-ozone treatment: an effective method for fine-tuning optical and electrical properties of suspended and substrate-supported MoS2" is of interest and is worth publishing after some minor revisions.

Below, you can find my comments:

1) In the introduction section, the connection between the state of the art and main purposes of the work is described quite weak. I suggest to improve the relationship between these two parts of the Introduction.

We would like to highlight here that in the second last paragraph of the introduction, we summarised the literature review into “….These promising studies are focused on the effect of UV-O3 treatment on the electrical properties of TMDs only. In 2D-TMD optoelectronic devices, the effect of UV-O3 treatment on their optical properties is also critical…..” and “…While the effect of UV-O3 on the electrical properties is consistent from one study to another, there is a contradiction about the effect of UV-O3 treatment on the optical properties of 2D-TMDs. More importantly, there is no systematic study on the effect of UV-O3 treatment on suspended, or multi-layer TMDs. Although monolayer (ML) TMDs have the strongest emission, the effect on few-layer TMDs is also important for the potential electronic and optoelectronic devices.….”.

To address these points, “..In this paper, we reveal that the effect of UV-O3 treatment does not only depend on the type of materials but also the number of layers and the substrate (the environment). To investigate the layer and substrate dependency of the UV-O3 treatment on the optical properties of semiconducting TMDs, we fabricated suspended and substrate-supported MoS2 samples with different numbers of layers from monolayer to 7 layers….”

2) Please explain why you took such time intervals for UV-O3 treatment: 2,4,6,8, 10 minutes. What is the scientific sence of these treatment time intevals. I didn't find any information in the manuscript.

We thank the reviewer for this comment. In the literature, there are few studies about using the UV-O3 treatment as a doping or surface functionalisation tool for 2D materials, as mentioned in the introduction sections (Ref: 21-29), most of which focused on controlling the free carrier density of the 2D materials and reducing the contact resistance. All work has kept the treatment time below 10 minutes to avoid material damage (Ref: 26, 29). We also observed a significant decrease in PL with 8 minutes of treatment time. In principle, we could use 1-minute intervals, but the change in PL was gentle, we do not expect to gain any further learning with a smaller time interval.

We have added a justification of the treatment time to the new version of the manuscript. “The main purpose of our study is to understand the layer and substrate dependency of the UV-O3 treatment time on the optical properties of the 2D TMDs, ideally maintaining their photoluminescence. We choose the UV-O3 treatment time of 1, 4, 6, and 8 mins on each sample.”

3) In Fig.2, the colors of the lines in the legends are very faded, so it is difficult to make out which curves they belong to. I recommend making the colors of the lines in the legends more vivid.

We thank the reviewer for pointing this out. The line sizes are doubled.

4) In Fig. 1b, it looks that the scales of the Si3N4 and SiO2 layer thicknesses (150 nm each) are not proportional to the scale of the pattern width (4 microns). I understand that this is a schematic, but it gives the reader the wrong impression at first glance. Please try to solve this problem.

We thank the reviewer for pointing this out. We have updated Figure 1.

5) In Fig 6, there is an error in the graph letter indexing:  "UV-O3 treatment time " corresponds to the d) and "applied drain-source voltage" corresponds to the c). You typed an inverse version.

We thank the reviewer for pointing this out. We have updated the figure caption.

Reviewer 2 Report

Comments and Suggestions for Authors

In the presented work entitled „Ultraviolet-ozone treatment: an effective method for fine-tuning optical and electrical properties of suspended and substrate-supported MoS2” by F. Sarcan et al., the Authors have investigated how the UV-O3 treatment affects the optical and electrical properties of 2D molybdenum disulfide (MoS2), with and without the presence of a dielectric substrate. The results showed that the influence of the UV-O3 treatment on photoluminescence depends on whether the substrate is present, as well as the number of layers. They also reveal that the substrate plays a critical role in the manipulation of the electrical and optical properties of MoS2, which should be considered in future device fabrication and applications.

In general, the article is written in a well-organized manner, and I did not notice any major errors in the conducted analysis. The discussion is clear and allows readers to get all the technical aspects of the presented investigations. The subject of the manuscript also appears to be timely, since it considers the 2D MoS2 material which is a representative example of the promising family of the transition metal dichalcogenide (TMDs) low-dimensional semiconductors. I summary, I believe that the presented results may be of interest to the wide scientific community. However, I believe there are several minor questions that should be answered before publication, in particular:

1.     A natural question that arises in terms of the modified electronic properties, is how the UV-ozone treatment influences the charge injection into the 2D MoS2. The 2D TMDs are known to exhibit Schottky barrier characteristics at the contact with a metal, with the so-called metal-induced gap states being responsible for their generation (ACS Appl. Mater. Interfaces 7 (2015) 25709 and Phys. Rev. B 97 (2018) 195315). Hence, can we expect, in terms of the mentioned previous results, some improvement of the charge injection (e.g. reduction of the Schottky barrier) in 2D MoS2 and overcome somewhat the Fermi level pinning due to the metal-induced gap states?

2.     Just a technical remark. The introduction sentence “In this paper, we reveal that the effect of UV-O3 treatment (…)” on page 2 should start new paragraph for better readability. See page 2.

3.     It seems to me that the Authors conduct theoretical part of their research without considering the spin-orbit coupling in 2D MoS2. However, the entire family of 2D TMDs is well-known for their strong spin-orbit coupling and the resulting effects that notably influences electronic properties (see for example the mentioned study on Schottky barriers in Phys. Rev. B 97 (2018) 195315). Obviously, the Authors can omit those effect, but should give reasoning why in their consideration such approximation can be done and how it influences their predictions (similarly how the Authors did it in case of the omitted substrate effects, see page 7).

Author Response

Reviewer 2.

In the presented work entitled „Ultraviolet-ozone treatment: an effective method for fine-tuning optical and electrical properties of suspended and substrate-supported MoS2” by F. Sarcan et al., the Authors have investigated how the UV-O3 treatment affects the optical and electrical properties of 2D molybdenum disulfide (MoS2), with and without the presence of a dielectric substrate. The results showed that the influence of the UV-O3 treatment on photoluminescence depends on whether the substrate is present, as well as the number of layers. They also reveal that the substrate plays a critical role in the manipulation of the electrical and optical properties of MoS2, which should be considered in future device fabrication and applications.

In general, the article is written in a well-organized manner, and I did not notice any major errors in the conducted analysis. The discussion is clear and allows readers to get all the technical aspects of the presented investigations. The subject of the manuscript also appears to be timely, since it considers the 2D MoS2 material which is a representative example of the promising family of the transition metal dichalcogenide (TMDs) low-dimensional semiconductors. I summary, I believe that the presented results may be of interest to the wide scientific community. However, I believe there are several minor questions that should be answered before publication, in particular:

  1. A natural question that arises in terms of the modified electronic properties, is how the UV-ozone treatment influences the charge injection into the 2D MoS2. The 2D TMDs are known to exhibit Schottky barrier characteristics at the contact with a metal, with the so-called metal-induced gap states being responsible for their generation (ACS Appl. Mater. Interfaces 7 (2015) 25709 and Phys. Rev. B 97 (2018) 195315). Hence, can we expect, in terms of the mentioned previous results, some improvement of the charge injection (e.g. reduction of the Schottky barrier) in 2D MoS2 and overcome somewhat the Fermi level pinning due to the metal-induced gap states?

We thank the reviewer for this thoughtful comment. There are some studies in the literature that focused on reducing the contact resistance between the 2D materials and the metal by controlling the carrier density of 2D materials. We have discussed this aspect in the introduction section: “   UV-O3 treatment is a useful process for a wide range of materials for surface modification [21]. It has been employed as an effective tool for defect engineering and doping in graphene and 2D TMDs. There are a few studies in the literature on UV-O3-induced p-type doping on graphene [22–24]. Liang et al. presented controllable p-type doping in a range of semiconducting TMDs (MoTe2, WSe2, MoSe2 and PtSe2) and proposed three mechanisms for UV-O3-induced hole doping: (1) charge transfer due to the interaction with oxygen molecules, (2) isoelectronic substitution of chalcogen atoms with oxygen atoms, and (3) charge transport over the oxide surface due to the transition metal oxides formation (MoO3, WO3 etc.) [20]. Zheng et al. showed that UV-O3 treatment is an effective method for p-type doping of MoTe2 field-effect transistors and it enhances its electrical performance enormously. The hole concentration and mobility are enhanced by nearly two orders of magnitude, and the conductivity by 5 orders of magnitude [25]. These promising studies are focused on the effect of UV-O3 treatment on the electrical properties of TMDs only.”

As the referee pointed out as well, the UV Ozone treatment possibly would modify the interaction with the metal contacts, and it would be interesting to conduct further investigation on top of the current literature, but this is beyond the scope of this work. Note here that in the transistor section of this work, we fabricated the transistor and then exposed the whole device to UV-Ozone, and in this way, we separate the effects from the metal contact and the UV-Ozone effects directly on the 2D material.

  1. Just a technical remark. The introduction sentence “In this paper, we reveal that the effect of UV-O3 treatment (…)” on page 2 should start new paragraph for better readability. See page 2.

We thank the reviewer for pointing this out. We have changed this.

  1. It seems to me that the Authors conduct theoretical part of their research without considering the spin-orbit coupling in 2D MoS2. However, the entire family of 2D TMDs is well-known for their strong spin-orbit coupling and the resulting effects that notably influences electronic properties (see for example the mentioned study on Schottky barriers in Phys. Rev. B 97 (2018) 195315). Obviously, the Authors can omit those effect, but should give reasoning why in their consideration such approximation can be done and how it influences their predictions (similarly how the Authors did it in case of the omitted substrate effects, see page 7).

We thank the reviewer for this comment. Spin-orbit coupling was not included in the calculations, which is a valid point. This can be justified in this case however, as the defect levels are shown to be >0.5 eV above the conduction band minimum (CBM), with spin-orbit coupling changing the bandgap far less than this value (for example, shown in our recent MoS2/TiS3 paper using PBE to shift indirect gap by ~0.1eV with no shift in the direct gap [37].This would mean that if SOC were included, the oxygen-induced defect levels would still be higher than the CBM, and hence unable to contribute to charge localisation – leaving the conclusions of this work unchanged.

To clarify this point, we have added a sentence in the experimental section of DFT in the updated version of our manuscript: “We note here that spin-orbit coupling was not included in any of these calculations, however, previous calculations on MoS2 have shown this to result in a 0.1eV shift in the band gap – significantly lower than the energy difference between the CBM and defect levels found in this work, hence would be unlikely to change any conclusions [37].”.